# Lignicolous Freshwater Fungi from Plateau Lakes in China (I): Morphological and Phylogenetic Analyses Reveal Eight Species of Lentitheciaceae, Including New Genus, New Species and New Records

**DOI:** 10.3390/jof9100962

**Published:** 2023-09-25

**Authors:** Hong-Wei Shen, Dan-Feng Bao, Saranyaphat Boonmee, Xi-Jun Su, Xing-Guo Tian, Kevin D. Hyde, Zong-Long Luo

**Affiliations:** 1College of Agriculture and Biological Science, Dali University, Dali 671003, China; hongweifungi@outlook.com (H.-W.S.); baodanfeng0922@gmail.com (D.-F.B.); suxijundali@163.com (X.-J.S.); 2Center of Excellence in Fungal Research, Mae Fah Luang University, Chiang Rai 57100, Thailand; saranyaphat.boo@mfu.ac.th (S.B.); 6271105511@lamduan.mfu.ac.th (X.-G.T.); kdhyde3@gmail.com (K.D.H.); 3School of Science, Mae Fah Luang University, Chiang Rai 57100, Thailand; 4Center for Yunnan Plateau Biological Resources Protection and Utilization, College of Biological Resource and Food Engineering, Qujing Normal University, Qujing 655011, China; 5School of Food and Pharmaceutical Engineering, Guizhou Institute of Technology, Guiyang 550003, China; 6Co-Innovation Center for Cangshan Mountain and Erhai Lake Integrated Protection and Green Development of Yunnan Province, Dali University, Dali 671003, China

**Keywords:** three new taxa, lignicolous freshwater fungi, plateau lake, phylogeny, taxonomy

## Abstract

During the investigation of lignicolous freshwater fungi in plateau lakes in Yunnan Province, China, eight Lentitheciaceae species were collected from five lakes viz. Luguhu, Qiluhu, Xingyunhu, Cibihu, and Xihu lake. Based on morphological characters and phylogenetic analysis of combined ITS, LSU, SSU, and *tef 1-α* sequence data, a new genus *Paralentithecium*, two new species (*Paralentithecium suae*, and *Setoseptoria suae*), three new records (*Halobyssothecium phragmitis*, *H. unicellulare,* and *Lentithecium yunnanensis*) and three known species viz. *Halobyssothecium aquifusiforme*, *Lentithecium pseudoclioninum,* and *Setoseptoria bambusae* are reported.

## 1. Introduction

Lignicolous freshwater fungi are those fungi that grow on submerged woody debris in freshwater habitats, including lentic (e.g., lakes, ponds, swamps, and pools), lotic (e.g., rivers, streams, creeks, brooks), and other habitats (e.g., cooling tower, tree holes) [1,2,3]. They play an important role in the material and energy cycle of freshwater ecosystems [4,5,6,7,8]. Lignicolous freshwater fungi are a highly diverse group, with the majority belonging to Dothideomycetes and Sordariomycetes (Ascomycota), and a few species in Eurotiomycetes and Orbiliomycetes [3,9,10,11,12]. Lignicolous freshwater fungi have been investigated worldwide, but mainly in lotic habitats of tropical, subtropical, and temperate regions [10,13,14,15,16], with a few from lentic habitats [17]. Those fungi in lentic habitats are poorly studied. This study collects submerged decaying wood from plateau lakes in Yunnan, China, to investigate the species diversity of lignicolous freshwater fungi in the lakes.

Yunnan Province is in the southwest of China, it is a low-latitude, high-altitude inland province, and is one of the biodiversity hotspots in the Yunnan–Guizhou Plateau [18]. Yunnan has three climatic zones, tropical (southwest, south, and southeast borders), subtropical (west, middle, and east), and temperate regions (high-elevation area in the northwest) [19]. The special geographical location and climatic conditions endow Yunnan with abundant natural resources. There are plateau cold-resistant biomes in the west and tropical biomes in the south and southwest. Plateau lakes are an important part of terrestrial lakes and an important part of regional water cycling. They are distributed at higher altitudes, have a large number, and have a wide basin area. They are sensitive to climate change and have made outstanding contributions to coping with global climate change and shaping regional biodiversity patterns [20,21]. There have been several biological studies conducted on plateau lakes in Yunnan, such as waterbirds [22,23], invasive fish [24], water plants [25,26,27], and microorganisms [2,17,28,29,30,31,32]. Yunnan has abundant lignicolous freshwater fungi resources, and from 1986 to 2021, a total of 281 lignicolous freshwater fungi taxa have been reported. These species were mainly reported in lotic habitats (rivers, streams), with only 53 (19%) species from plateau lakes [12].

Species of Lentitheciaceae from freshwater habitats are mainly in *Halobyssothecium*, *Lentithecium*, *Setoseptoria,* and *Tingoldiago* [14,15,33,34,35,36,37]. The family was introduced by Zhang et al. [35] to accommodate those lentitheciaceous taxa that have narrow peridia, fusiform to broadly cylindrical pseudoparaphyses, hyaline ascospores with 1–3-transverse septa and containing refractive globules, surrounded by a mucilaginous sheath or extended appendage-like sheaths and asexual morphs are stagonospora-like or dendrophoma-like [14,38,39,40]. Currently, more than 100 species are reported in Lentitheciaceae. The last treatment of Lentitheciaceae was provided by Wijayawardene et al. [41] with acceptance of 18 genera: *Crassoascoma* [42], *Darksidea* [43], *Groenewaldia* [44], *Halobyssothecium* [45], *Katumotoa* [46], *Keissleriella* [47], *Lentithecium* [35], *Murilentithecium* [40], *Neolentithecia* [48], *Neoophiosphaerella* [34], *Phragmocamarosporium* [49], *Pleurophoma* [50,51], *Poaceascoma* [52], *Pseudokeissleriella* [53], *Pseudomurilentithecium* [54], *Setoseptoria* [37], *Tingoldiago* [55], and *Towyspora* [56].

We are investigating the diversity of lignicolous freshwater fungi from plateau lakes in Yunnan Province, and 13 collections of lentitheciaceous-like taxa were obtained. Based on morphological and multigene phylogenetic analysis, a new genus *Paralentithecium* is introduced to accommodate *P. aquaticum,* and a new taxon *P. suae*, *Setoseptoria suae* sp. nov. and new records *Halobyssothecium* and *Lentithecium* are described and illustrated. The sexual morph of *Halobyssothecium phragmitis* is also introduced.

## 2. Materials and Methods

### 2.1. Samples Collection

The fresh samples were submerged in lake water with a diameter of less than 2 cm and a length of more than 20 cm, including tree trunks, branches, twigs, and rotten branches of grasses. The specimens in this study were collected from Dali City (Cibihu, Xihu, and Erhai Lakes), Lijiang City (Luguhu Lake), and Yuxi City (Xingyunhu and Qiluhu Lakes) in Yunnan. The collected samples were placed in plastic ziplock bags and were taken back to the laboratory for processing.

### 2.2. Sample Processing and Cultivation

The samples were brought to the laboratory in ziplock bags to avoid moisture loss and then trimmed to 15 cm in length with pruning scissors. Each sample with a label number that is attached to the end of the sample with a thumbtack (Figure 1a). In addition, plastic boxes with the size of 24 cm × 16 cm × 6 cm were prepared. First, rinse the inside of the plastic box with sterile water, then wipe the entire plastic box with 75% alcohol. After drying, put two layers of sterilized tissues on the bottom of the box, lay three sterilized straws on the tissues to prevent the sample from directly touching the sterilized tissues, and add an appropriate amount of sterile water (the water soaks sterile tissues, but accumulates at the bottom), and then arrange the processed samples horizontally on the straw, ten samples in each plastic box, and label the boxes in obvious places (Figure 1b,c). The samples were placed on a culture rack and incubated at room temperature for one week (Figure 1d).

### 2.3. Morphological Studies and Isolation

Macromorphological characters of samples were observed using Optec SZ 760 compound stereomicroscope (Chongqing Optec Instrument Co., Ltd., Chongqing, China). The temporarily prepared microscope slide was placed under a Nikon ECLIPSE Ni-U compound stereomicroscope (Nikon, Tokyo, Japan) for observation and microscopic morphological photography. The morphology of colonies on native substrates was photographed with a Nikon SMZ1000 stereo-zoom microscope. Indian ink was used to reveal the presence of a gelatinous sheath around the ascospores or conidia. The measurements of photomicrographs were obtained using Tarosoft (R) Image Frame Work version 0.9.7. Images were edited with Adobe Photoshop CS5 Extended version 12.0.0.0 software (Adobe Systems, San Jose, CA, USA).

Single spore isolations were performed as follows: the tip of a sterile toothpick dipped in sterile water was used to capture the conidia of the target colony directly from the specimen; the conidia were then streaked on the surface of water agar (WA, Composition: Agar 20 g/L, Chloramphenicol 0.1 g/L) or potato dextrose agar (PDA, CM123, Composition: Potato infusion 5.0 g/L, Dextrose 20 g/L, Agar 20 g/L, Chloramphenicol 0.1 g/L, from Beijing Bridge Technology Co., Ltd., Beijing, China) and incubated at room temperature overnight. The single germinated conidia were transferred to fresh PDA medium and incubated at room temperature. A few of the remaining germinated spores in the media plate were separated along with agar using a needle and transferred onto water-mounted glass slides for photographs to capture the germination position of the germ tubes.

After finalizing the observation and isolation, the specimens were dried under natural light, wrapped in absorbent paper, and placed in a ziplock bag with mothballs. Specimens were deposited in the herbarium of Kunming Institute of Botany, Academia Sinica (KUN-HKAS). The living cultures were deposited in the China General Microbiological Culture Collection Center (CGMCC) and Kunming Institute of Botany Culture Collection (KUNCC). MycoBank numbers are registered in the MycoBank database (https://www.mycobank.org/Registration%20home (accessed on 4 August 2023)). Entries will be added to the Greater Mekong Subregion database [57].

### 2.4. DNA Extraction, PCR Amplification and Sequencing

DNA extraction, PCR amplification, sequencing, and phylogenetic analysis were done following the methods of Dissanayake et al. [58]. Mycelia for DNA extraction from each isolate was grown on PDA for 3–4 weeks at room temperature. Total genomic DNA was extracted from 100 to 300 mg axenic mycelium via scraping from the edges of the growing culture using a sterile scalpel and transferred to a 1.5 mL microcentrifuge tube using sterilized inoculum needles. The mycelium was ground to a fine powder with liquid nitrogen or quartz sand to break the cells for DNA extraction. When the cultures could not be maintained with some of the collected samples, fruiting structures (20–50 mg) were removed from the natural substrate using a sterile scalpel placed on sterile paper and then transferred to a 1.5 mL microcentrifuge tube. DNA was extracted with the TreliefTM Plant Genomic DNA Kit (TSP101) following the manufacturer’s guidelines.

Four gene regions, ITS, LSU, SSU, and *tef 1-α* were amplified using ITS5/ITS4 [59], LR0R/LR5 [60], NS1/NS4 [59], and EF1-983F/EF1-2218R [61] primer pairs, respectively. The PCR mixture contained 12.5 µL of 2× Power Taq PCR MasterMix (a premix and ready-to-use solution, including 0.1 Units/µL Taq DNA Polymerase, 500 µm dNTP Mixture each (dATP, dCTP, dGTP, dTTP), 20 mm Tris–HCl pH 8.3, 100 Mm KCl, 3 mM MgCl_2_, stabilizer, and enhancer), 1 µL of each primer including forwarding primer and reverse primer (10 µm), 1 µL template DNA extract and 9.5 µL deionized water. The PCR thermal cycling conditions of ITS and SSU were as follows: 94 °C for s min, followed by 35 cycles of denaturation at 94 °C for 30 s, annealing at 56 °C for 50 s, elongation at 72 °C for 1 min, and a final extension at 72 °C for 10 min; LSU and *tef 1-α* were as follows: 94 °C for 3 min, followed by 35 cycles of denaturation at 94 °C for 30 s, annealing at 55 °C for 50 s, elongation at 72 °C for 1 min, and a final extension at 72 °C for 10 min. PCR products were then purified using minicolumns, purification resin, and buffer according to the manufacturer’s protocols (Amersham product code: 27-9602-01). The sequences were carried out at Beijing Tsingke Biological Engineering Technology and Services Co., Ltd. (Beijing, China).

### 2.5. Phylogenetic Analyses

ITS, LSU, SSU, and *tef 1-α* sequence data used for phylogenetic analysis are selected based on the preliminary identification results and the related publications [14,15]. The sequences were aligned using MAFFT online service: multiple alignment program MAFFT v.7 (http://mafft.cbrc.jp/alignment/server/index.html (accessed on 30 August 2023)) [62], and sequence trimming was performed with trimAl v1.2 for Windows, and all parameters were set by default (http://trimal.cgenomics.org for specific operation steps (accessed on 30 August 2023)) [63]. The sequence dataset was combined using SquenceMatrix v.1.7.8 [64]. FASTA alignment formats were changed to PHYLIP and NEXUS formats by the website: ALignment Transformation EnviRonment (ALTER) (http://sing.ei.uvigo.es/ALTER/ (accessed on 30 August 2023)) [65]. The alignments and phylogenetic trees were deposited in TreeBASE (http://www.treebase.org/ (accessed on 31 August 2023), accession number: 30729-30733).

The single-gene phylogenetic tree was obtained based on maximum likelihood (ML) only, and the multigene phylogenetic tree was obtained based on maximum likelihood (ML) and Bayesian criterion (BI). ML tree and BI tree were run on the CIPRES Science Gateway portal [66,67,68,69]. MrModeltest v. 2.3 [70] was run under the AIC (Akaike Information Criterion) implemented in PAUP v. 4.0b10. to evaluate the best-fit model in both ML and BI analyses. ML analyses for the datasets were performed with RAxML-HPC2 on XSEDE v. 8.2.10 [66] using the determined best-fit substitution model with 1000 bootstrap iterations. The BI analysis was computed with MrBayes v. 3.2.6 [69]. Six simultaneous Markov chains were run with a suitable number of generations, and trees were sampled every 100th generation, ending the run automatically when the standard deviation of split frequencies dropped below 0.01. Alignment gaps were treated as missing characters in the analysis of the combined data set, where they will occur in relatively conserved regions. Trees were inferred using the heuristic search option with 1000 random sequence additions, with maxtrees set at 1000. Phylogenetic trees were visualized using FigTree v1.4.0 (http://tree.bio.ed.ac.uk/software/figtree/ (accessed on 31 August 2023)), editing and typesetting using Adobe Illustrator (AI) (Adobe Systems Inc., San Jose, CA, USA). The new sequences were submitted in GenBank, and the strain information used in this paper is provided in Table 1.

## 3. Results

### 3.1. Phylogenetic Analysis

The combined ITS, LSU, SSU, and *tef 1-α* dataset comprises 147 taxa, including nine genera of Lentitheciaceae, with *Pleomonodictys capensis* (CBS 968.97) and *P. descalsii* (CBS 142298) as outgroup taxa (Figure 2). The dataset comprised 3777 characters (LSU: 1285 bp; SSU: 1021 bp; ITS: 539 bp; *tef 1-α*: 932 bp, including gaps). Maximum likelihood (ML) analysis and Bayesian analysis produced similar topologies that were consistent across the major clades. The likelihood of the final tree is evaluated and optimized under GAMMA. The best RAxML tree with a final likelihood value of −31,318.755060 is presented (Figure 2). The matrix had 1636 distinct alignment patterns, with 27.52% undetermined characters or gaps. Estimated base frequencies were as follows: A = 0.239720, C = 0.248808, G = 0.272414, T = 0.239058; substitution rates AC = 1.212047, AG = 2.534776, AT = 1.388124, CG = 1.249521, CT = 7.002685, GT = 1.000000, α = 0.226056, Tree-Length: 3.286461. Bayesian analyses generated 4412 trees (average standard deviation of split frequencies: 0.009960) from which 3310 were sampled after 25% of the trees were discarded as burn-in. The alignment contained a total of 1441 unique site patterns. Bootstrap support values with an ML greater than 75%, and Bayesian posterior probabilities (PP) greater than 0.97 are given above the nodes.

The multigene phylogenetic analyses showed that the 13 fresh collections clustered within Lentitheciaceae. Five known species, *Halobyssothecium aquifusiforme* (KUNCC 22-12665), *H. phragmitis* (KUN-HKAS 127181), *H. unicellulare* (KUNCC 22-12413), *Lentithecium pseudoclioninum* (KUNCC 22-12414 and KUNCC 22-12415), *L. yunnanensis* (KUNCC 22-124201, KUNCC 22-12421 and KUNCC 22-12422) and *Setoseptoria bambusae* (KUNCC 22-12416, KUNCC 22-12417 and KUNCC 22-12418) clustered with their ex-type strains, respectively. *Paralentithecium suae* (KUNCC 22-12412) clustered sister to *P. aquaticum* (CBS 123099) in an independent clade within Lentitheciaceae. *Setoseptoria suae* (KUNCC 22-12419) was placed sister to *S. phragmitis* (CBS 114804 and CBS 114966). Single-gene phylogenies are shown as Appendix A) because they resulted in being less informative and resolutive than those based on the four-loci concatenated tree.

### 3.2. Taxonomy

*Halobyssothecium aquifusiforme* J. Yang, Jian K. Liu & K.D. Hyde, Fungal Diversity 119: 39 (2023). Figure 3.

Index Fungorum number: IF559450; Facesoffungi number: FoF12783.

*Saprobic* on submerged decaying wood in a freshwater lake. Asexual morph: Undetermined. Sexual morph: *Ascomata* 354–382 µm high, 328–366 µm wide, immersed, clustered, sometimes solitary, scattered, subglobose or ellipsoidal, dark brown to black, carbonaceous, uniloculate, ostiolate. *Ostiolar* neck central, 86–114 µm long, 138–168 µm wide, papillate, rounded, short, dark brown, composed of several layers of pseudoparenchymatous cells. *Peridium* 22–35 µm thick, composed of several layers of pseudoparenchymatous cells, an outer layer composed of black cells, arranged in a *textura angularis*, inner layer composed of hyaline, flattened cells, arranged in a *textura angularis*. *Pseudoparaphyses* about 2 µm wide, branched, septate, hyaline, filamentous, anastomosing above the asci. *Asci* 97–129 × 13–16 µm ($\bar{x}$ = 113 × 14 µm, *n* = 20) µm, 8-spored, clavate to subcylindrical, bitunicate, fissitunicate, apex rounded, short pedicellate, with an ocular chamber. *Ascospores* (20–)24–27 × 7–8 µm ($\bar{x}$ = 25 × 8 µm, *n* = 40), overlapping, uniseriate to biseriate, central cells are brown to dark brown, 1-septate when young, 3-septate when mature, constricted at the septa, slightly curved, fusiform, guttulate, conical and narrowly rounded at the ends, one cell on the central septum side is swollen, lacking gelatinous sheaths or appendages.

Culture characteristics: Ascospore germinating on PDA within 12 h. Colonies on PDA reaching 3 cm diameter in 6 weeks at room temperature. Mycelium superficial, initially white, later becoming brown to black, with pale brown dense aerial mycelium on the surface, mastoid, marginal mycelium smooth, sparse, brown to black; from below, light brown at the center, dark brown at the margin.

Material examined: China, Yunnan Province, Dali City, Eryuan County, Cibihu Lake, 26°09′59″ N, 99°55′27″ E (2050 m), on unknown submerged decaying wood, 21 July 2021, S.P. Huang and L.L. Li, L788 (KUN-HKAS 124599), living cultures (KUNCC 22-12665).

Known host and distribution: China, Guizhou Province, Anshun City, Gaodang village, 26.071° N, 105.698° E, Suoluo River, on decaying wood submerged in a freshwater stream HKAS 112638, (holotype), HKAS 112641 (paratype) [15].

Notes: The phylogenetic analysis showed that our new strain, KUNCC 22-12665 clustered sister to *Halobyssothecium aquifusiforme* (GZCC 20-0481 and MFLUCC 19-0305) with 99% ML/1.00 PP supports (Figure 2). Our species is similar to *H. aquifusiforme* in having immersed, subglobose ascomata, and fusiform, guttulate, septate ascospores which are constricted at the septum [15]. We, therefore, identified our new collection as *H. aquifusiforme* and provided detailed descriptions and illustrations for it. *Halobyssothecium aquifusiforme* is an aquatic species that was collected on submerged decaying wood in a freshwater stream in Guizhou, China. Our two new collections were collected from a plateau lake in Yunnan.

*Halobyssothecium phragmitis* M.S. Calabon, E.B.G. Jones, S. Tibell & K.D. Hyde, Mycological Progress 20: 711 (2021). Figure 4.

Index Fungorum number: IF558090; Facesoffungi number: FoF 09431.

*Saprobic* on submerged decaying wood in a freshwater lake. Sexual morph: *Ascomata* 529–566 µm high, 545–691 µm wide, immersed or semi-immersed, solitary to gregarious, scattered, subglobose or ellipsoidal, dark brown, subcarbonaceous or coriaceous, uniloculate, with indistinct ostiolate. *Ostiolar neck* 172–265 µm high, 184–320 µm wide, central, papillate, rounded, short, dark brown, composed of several layers of pseudoparenchymatous cells. *Peridium* 26–77 µm thick, composed of several layers of pseudoparenchymatous cells, an outer layer composed of brown cells, arranged in a *textura angularis* and *textura globulosa*, and an inner layer composed of hyaline, flattened cells, arranged in a *textura angularis. Pseudoparaphyses* 2–3 µm wide, septate, hyaline, filiform, branched, anastomosing above the asci. *Asci* (102–)111–130(–137) × 10–12 µm ($\bar{x}$ = 121 × 11 µm, *n* = 30), 8-spored, clavate to subcylindrical, bitunicate, fissitunicate, short pedicellate with an ocular chamber. *Ascospores* 22–27 × 5–6 µm ($\bar{x}$ = 25 × 6 µm, *n* = 40), overlapping, uniseriate to biseriate, fusiform with narrow ends, cells swollen nearly central septum and gradually narrow toward ends, slightly curved, pale brown to dark brown and lightening from central cells to the end cells, 1-septate when young, 5-septate when mature, and constricted at the septa, lacking gelatinous sheaths or appendages. Asexual morph: Coelomycetes [14].

Material examined: China, Yunnan Province, Dali City, Eryuan County, Xihu Lake, 26°00′33″ N, 100°03′35″ E (1970 m), on unknown submerged decaying wood, 8 May 2021, S.P. Huang and L.L Li, L783 (KUN-HKAS 127181).

Known host and distribution: SWEDEN, Gotland, Kappelshamnsviken, on dead *Phragmites culm* (Poaceae), MFLU 20–0550 (holotype); ibid., Sudersand, on dead *Phragmites* (Poaceae) stem, MFLU 20–0552 (paratype) [14].

Notes: *Halobyssothecium phragmitis* was introduced by Calabon et al. [14], and only the asexual morph is known. This species was collected on *Phragmites* (*Poaceae*) culm in Europe. Phylogenetic analysis combined with ITS, LSU, SSU, and *tef 1-α* sequence data showed that our new collection (KUN-HKAS 127181) clustered with two strains of *H. phragmitis* (MFLUCC 20–0223 and MFLUCC 20–0225). The comparison of ITS, LSU, SSU, and *tef 1-α* sequences between our new collection (KUN-HKAS 127181) and the ex-type of *H. phragmitis* (MFLUCC 20–0226) showed 8 bp, 1 bp, 3 bp, and 3 bp differences, respectively. Morphologically, our new collection is similar to other sexual members of *Halobyssothecium* in having immersed or semi-immersed, subglobose or ellipsoidal, dark brown, subcarbonaceous or coriaceous ascomata, clavate to subcylindrical, bitunicate asci and 3-septate, fusiform ascospores [14,15,45,71]. Based on phylogenetic analysis and morphological evidence, we identified our new collection as *H. phragmitis*, and described its asexual morph. This is the first report of this species in China [14].

*Halobyssothecium unicellulare* (Abdel-Aziz) M.S. Calabon, K.D. Hyde & E.B.G. Jones, Mycological Progress 20: 715 (2021). Figure 5.

Index Fungorum number: IF558094; Facesoffungi number: FoF 09437

*Saprobic* on submerged decaying wood in a freshwater lake. Sexual morph: Undetermined. Asexual morph: Coelomycetes. *Conidiomata* 135–178 µm high, 205–242 µm wide, immersed to semi-immersed, most immersed, clustered, sometimes solitary, scattered, subglobose or ellipsoidal, uniloculate, dark brown to black, carbonaceous, short ostiolate, papillate, rounded. *Conidiomatal walls* 14–31 µm thick, composed of several layers of hyaline to black–brown cells of *textura angularis*. *Conidiophores* are reduced to conidiogenous cells. *Conidiogenous cells* 5–12 × 3–5 µm ($\bar{x}$ = 8 × 4 µm, *n* = 30), hyaline, thin-walled, holoblastic, smooth, subglobose to pear-shaped, swollen at the base, sometimes one conidiogenous cell producing two conidia. *Conidia* 9–11 × 4–5 µm ($\bar{x}$ = 10 × 5 µm, *n* = 60), subglobose, ovate, clavate, ellipsoid, allantoid or irregular, hyaline, aseptate, several small to one big guttulate, smooth-walled.

Culture characteristics: Conidia germinating on PDA within 12 h and germ tubes produced from one end of the conidia. Colonies on PDA, circular, reaching 5 cm in one month at room temperature, flat surface, pale brown to brown in PDA medium. Mycelium superficial, white to brown, hairy, effuse with wavy edge, dense, circular, raised, undulate to filiform with age; reverse light brown in the middle, with a dark brown deposit on the outside.

Material examined: China, Yunnan Province, Dali City, Eryuan County, Xihu Lake, 26°00′33″ N, 100°03′35″ E (1970 m), on unknown submerged decaying wood, 8 May 2021, S.P. Huang and L.L Li, L412 (KUN-HKAS 124589), living cultures, KUNCC 22-12413.

Known host and distribution: EGYPT, Sohag City, on decayed wood submerged in the River Nile, CBS H-22674 (holotype) [72].

Notes: The multigene phylogenetic analysis showed that our new collection (KUNCC 22-12413) clustered with the ex-type strain of *Halobyssothecium unicellulare* (MD 6004) with 91% ML/1.00 PP support (Figure 2). Morphologically, our new collection fits well with the original description of *H. unicellulare* [72]. The nucleotide comparison of LSU and SSU sequence data between our new collection (KUNCC 22-12413) and *H*. *unicellulare* (MD 6004) revealed 2 bp (including one gap) and 1 bp (including one gap) differences, respectively. We therefore identified it as *H. unicellulare* and it was reported from China for the first time.

*Lentithecium pseudoclioninum* Kaz. Tanaka & K. Hiray, Studies in Mycology 82: 99 (2015). Figure 6.

Index Fungorum number: IF811309; Facesoffungi number: FoF12785.

*Saprobic* on submerged decaying wood in a freshwater lake. Asexual morph: Undetermined. Sexual morph: *Ascomata* 201–310 µm high, 227–274 µm wide, black, semi-immersed, gregarious, erumpent, globose or subglobose, uniloculate, ostiolate. *Ostiolar neck* central, papillate, 92–110 µm long, 100–107 µm wide. *Peridium* 20–32 µm, thick-walled, brown to dark brown cells, composing several layers of pseudoparenchymatous cells of *textura angularis*, outer layers heavily pigmented, inner layers hyaline to pale brown, flattened. *Pseudoparaphyses* 2–3 µm wide, filamentous, branched septate. *Asci* 98–118 × 14–16 µm ($\bar{x}$ = 108 × 15 µm, *n* = 30), 8-spored, bitunicate, fissitunicate, cylindric-clavate, slightly curved, pedicellate, apex rounded with a minute ocular chamber. *Ascospores* 28–32 × 8–10 µm ($\bar{x}$ = 30 × 9 µm, *n* = 30), overlapping uni- to biseriate, narrowly fusiform, with a nearly median primary septum, constricted at the septum, hyaline, guttulate, usually with 2–4 larger guttules, asymmetrical, broadly fusiform, narrowly rounded at the ends, with a mucilaginous sheath.

Culture characteristics: Ascospore germinating on PDA within 12 h and germ tubes produced from the ends of the spore. Colonies on PDA, circular, reaching 5 cm in one month at room temperature, smooth surface, papillae, brown to dark brown. Mycelium superficial, hairy, smooth, circular, reverse grayish; reverse pale to brown, crack at the middle, flocculent at the edge.

Material examined: China, Yunnan Province, Dali City, Eryuan County, Xihu Lake, 26°00′33″ N, 100°03′35″ E (1970 m), on unknown submerged decaying wood, 8 May 2021, S.P. Huang and L.L Li, L413 (KUN-HKAS 124590), living cultures (KUNCC 22-12414); ibid., Erhai Lake, 26°00′32″ N, 100°03′35″ E (1970 m), on unknown submerged decaying wood, 01 April 2021, Z.Q. Zhang, L445 (KUN-HKAS 124593), living cultures (KUNCC 22-12415)

Known host and distribution: JAPAN, Aomori, Hirosaki, Aoki, Mohei pond, on submerged twigs of woody plant, KT 1113 (holotype) and KT 1111 (paratype); China, Guizhou Province, Weining City, Caohai National Nature Reserve, near 26.817° N, 104.217° E, on submerged decaying aquatic plants in Caohai lake, GZAAS 20-0378 [15].

Notes: Our two new collections are morphologically consistent with the holotype of *Lentithecium pseudoclioninum* [34]. In addition, phylogenetic analysis revealed that these two collections clustered with *L. pseudoclioninum* (Figure 2). Based on morphological and phylogenetic evidence, we identified our new collection as *L. pseudoclioninum*. *Lentithecium pseudoclioninum* has been collected on submerged twigs of woody plants in China and Japan [15,34]. Our two specimens were collected from a freshwater plateau lake in Yunnan, China.

*Lentithecium yunnanensis* W.H. Lu, Karun. & Tibpromma, Phytotaxa 554: 108 (2022). Figure 7.

Index Fungorum number: IF559622; Facesoffungi number: FoF 10778.

*Saprobic* on submerged decaying wood in a freshwater lake. Asexual morph: Undetermined. Sexual morph: *Ascomata* 246–285 µm high, 179–229 µm wide, immersed to semi-immersed, clustered, sometimes solitary, scattered, subglobose or ellipsoidal, dark brown to black, carbonaceous, uni- to bi-loculate, with indistinct ostiolate. *Ostiolar neck* central, papillate, 127–156 µm long, 96–110 µm wide. *Peridium* 11–21 µm thick, composed of several layers of pseudoparenchymatous cells, outer layer composed of back brown to brown cells, arranged in *textura angularis*, inner layer composed of hyaline cells, arranged in *textura angularis*. *Pseudoparaphyses* about 2 µm wide, hyaline, filamentous, branched, septate, globose to subglobose swollen at the apex, sometimes swollen at the septum, anastomosing at the apex, embedded in a hyaline gelatinous matrix. *Asci* 98–117 × 14–15 µm ($\bar{x}$ = 108 × 15 µm, *n* = 15), 8-spored, clavate to subcylindrical, bitunicate, apex rounded, short pedicellate with an ocular chamber. *Ascospores* 27–30 × 5–6 µm ($\bar{x}$ = 28 × 6 µm, *n* = 30), overlapping, uniseriate to biseriate, hyaline, 1-septate, smooth, constricted at the septa, slightly curved, guttulate, lacking gelatinous sheaths or appendages.

Culture characteristics: Ascospore germinating on PDA within 12 h and germ tubes produced from both ends of the spore. Colonies on PDA, circular, reaching 6 cm in 45 days at room temperature, smooth surface, papillae, brown in PDA medium. Mycelium superficial, brown to dark brown, hairy, smooth, circular; reverse brown to dark brown, crack at the middle, flocculent at the edge.

Material examined: China, Yunnan Province, Dali City, Eryuan County, Xihu Lake, 26°17′37″ N, 99°58′33″ E (2100 m), on unknown submerged decaying wood, 22 July 2021, L.L. Li, L680 (KUN-HKAS 124598), living cultures (KUNCC 22-12420 = KUNCC 22-12422); ibid., 26°17′24″ N, 99°57′56″ E (2100 m), on unknown submerged decaying wood, 22 July 2021, X.J. Yuan, L679 (KUN-HKAS 124597), living culture (KUNCC 22-12421).

Known host and distribution: China, Yunnan, Kunming, Songhua Dam Reservoir, on dead culms of *Artemisia* sp., HKAS 123192 (holotype) [73].

Notes: *Lentithecium yunnanensis* is a terrestrial species introduced by Lu et al. [73] that occurs on dead culms of *Artemisia* sp. near humid places. We collected two *Lentithecium*-like collections from decaying wood submerged in Xihu Lake, Dali, Yunnan Province. Phylogenetic analysis showed that our two new collections clustered with two strains of *L. yunnanensis* (KUNCC 22-10776 and KUNCC 22-10776). In addition, the morphology of our two collections is similar to the holotype of *L. yunnanensis* in having semi-immersed to immersed, subglobose to globose ascomata with short ostioles, and hyaline, clavate to fusiform, septate ascospores. Therefore, the two new collections were identified as *L. yunnanensis*, which was reported from the freshwater habitat for the first time.

*Paralentithecium* H.W. Shen, K.D. Hyde & Z.L. Luo gen. nov.

MycoBank number: 849738.

Etymology: referring to the comparable morphological characters to that of *Lentithecium*.

*Saprobic* on submerged decaying wood in a freshwater lake. Asexual morph: Undetermined. Sexual morph: *Ascomata* immersed to semi-immersed, clustered, sometimes solitary, scattered, subglobose or ellipsoidal, dark brown to black, carbonaceous, uni- to bi-loculate, with indistinct ostiolate. *Peridium* thick, composed of several layers of pseudoparenchymatous cells, an outer layer composed of back brown to brown cells, arranged in *textura angularis*, and an inner layer composed of hyaline cells, arranged in *textura angularis*. *Pseudoparaphyses* thick, hyaline, filamentous, branched, septate, globose to subglobose swollen at the apex, sometimes swollen at the septum, anastomosing at the apex, embedded in a hyaline gelatinous matrix. *Asci* 8-spored, clavate to subcylindrical, bitunicate, apex rounded, short pedicellate with an ocular chamber. *Ascospores* overlapping, uniseriate to biseriate, hyaline, 1-septate, smooth, constricted at the septa, slightly curved, with gelatinous sheaths. 

Type species: *Paralentithecium aquaticum* (Yin. Zhang, J. Fourn. & K.D. Hyde) H.W. Shen & Z.L. Luo.

*Paralentithecium aquaticum* (Yin. Zhang, J. Fourn. & K.D. Hyde) H.W. Shen & Z.L. Luo, comb. nov.

MycoBank number: MB 512791.

Basionym: *Lentithecium aquaticum* Yin. Zhang, J. Fourn. & K.D. Hyde, Fungal Diversity 38: 234 (2009).

Known host and distribution: FRANCE, Ariège, Rimont, Peyrau, on submerged wood of *Fraxinus excelsior*; on submerged wood of *Alnus glutinosa*; Le Baup brook, along D 18, on submerged wood of *Platanus* sp. [35].

Notes: *Lentithecium aquaticum* was introduced by Zhang et al. [35] based on phylogenetic analysis and morphological characteristics. The placement of this species was not stable and has been changed by several studies [10,45,71,74]. Previous phylogenetic analyses indicated that *Lentithecium aquaticum* did not cluster with other *Lentithecium* species, and it formed an individual lineage basal to *Darksidea*, *Halobyssothecium* and *Lentithecium* [10,14,34,41,48,71]. Furthermore, phylogenetic studies of Dayarathne et al. [45] and Devadatha et al. [71] showed that *L. aquaticum* clustered within *Setoseptoria*. Several other studies excluded *L. aquaticum* from *Lentithecium* [10,74]. The latest phylogenetic analysis based on combined ITS, LSU, SSU, and *tef 1-α* genes showed that *L. aquaticum* formed a separate lineage outside of *Lentithecium* [48]. Our phylogenetic analysis shows that *L. aquaticum* clusters with our new collection KUNCC 22-12412 and forms a distinct lineage within Lentitheciaceae with 100 ML/1.00 PP support (Figure 2). Therefore, we propose a new genus, *Paralentithecium* to accommodate *Paralentithecium aquaticum* (*Lentithecium aquaticum*) and a new species *P. suae*.

*Paralentithecium suae* H.W. Shen, K.D. Hyde & Z.L. Luo sp. nov. Figure 8.

MycoBank number: 849739; Facesoffungi number: FoF 14876.

Etymology: “suae” (Lat.) in memory of the Chinese mycologist Prof. Hong-Yan Su (4 April 1967–3 May 2022).

Holotype: KUN-HKAS 124587.

*Saprobic* on submerged decaying wood in a freshwater lake. Asexual morph: Undetermined. Sexual morph: *Ascomata* 212–253 µm high, 175–204 µm wide, immersed to semi-immersed, clustered, sometimes solitary, scattered, subglobose or ellipsoidal, dark brown to black, carbonaceous, uni- to bi-loculate, with indistinct ostiolate. *Peridium* 17–32 µm thick, composed of several layers of pseudoparenchymatous cells, outer layer composed of bark brown to brown cells, arranged in *textura angularis*, inner layer composed of hyaline cells, arranged in *textura angularis*. *Pseudoparaphyses* 2–3 µm wide, hyaline, filamentous, branched, septate, globose to subglobose swollen at the apex, sometimes swollen at the septum (6–10 µm wide), anastomosing at the apex, embedded in a hyaline gelatinous matrix. *Asci* 104–134 × 24–28 µm ($\bar{x}$ = 119 × 26 µm, *n* = 25), 8-spored, clavate to subcylindrical, bitunicate, apex rounded, short pedicellate with an ocular chamber. *Ascospores* 28–34 × 11–14 µm ($\bar{x}$ = 31 × 13 µm, *n* = 40), overlapping, uniseriate to biseriate, hyaline, 1-septate, broadly fusiform, smooth, constricted at the septa, slightly curved, guttulate, with gelatinous sheaths.

Culture characteristics: Ascospore germinating on PDA within 12 h and germ tubes produced from both ends of the spore. Colonies on PDA, circular, reaching 4–5 cm in one month at room temperature, smooth surface, papillae, brown to dark brown, olive green in PDA medium. Mycelium superficial, brown to dark brown, hairy, smooth, circular; reverse dark brown, crack at the middle, flocculent at the edge, dark brown with greenish.

Material examined: China, Yunnan Province, Lijiang City, Ninglang County, Luguhu Lake, 27°44′15″ N, 100°45′16″ E (2700 m), on unknown submerged decaying wood, 5 March 2021, Z.Q. Zhang and L. Sha, L184 (KUN-HKAS 124587, holotype), ex-type living cultures (CGMCC 3.24265 = KUNCC 22–12412).

Notes: In our phylogenetic analysis, *Paralentithecium suae* clustered with *P. aquaticum* with 100% ML/1.00 PP support (Figure 2). Comparison of ITS, LSU, SSU, and *tef 1-α* sequences between *Paralentithecium suae* and *P. aquaticum* revealed 11 bp, 4 bp, 4 bp, and 22 bp differences, respectively. *Paralentithecium suae* resembles *P*. *aquaticum* in having hyaline, 1-septate, broadly fusiform ascospores with gelatinous sheaths [35]. However, *P. suae* is distinct from *P. aquaticum* in having globose to subglobose pseudoparaphyses that are swollen at the apex and sometimes swollen at the septum. In contrast, the pseudoparaphyses of *P. aquaticum* are not swollen. In addition, ascospores of *P. aquaticum* contain four refractive globules, while *P. suae* has ascospores with many small guttules [35]. Therefore, we introduce *P. suae* as a new species.

*Setoseptoria bambusae* J. Yang, Jian K. Liu & K.D. Hyde, Fungal Diversity 119: 44 (2022). Figure 9.

Index Fungorum number: IF559452; Facesoffungi number: FoF12786.

*Saprobic* on submerged decaying wood in a freshwater lake. Asexual morph: Undetermined. Sexual morph: *Ascomata* 245–375 µm high, 194–296 µm wide, black, superficial to semi-immersed, gregarious, fully or partly erumpent, globose, uniloculate, ostiolate. *Peridium* 26–39 µm wide, thick, multi-layered, outer layer most heavily pigmented, comprising blackish to dark brown amorphous layer, middle layer heavily pigmented, inner layer, pale brown to hyaline, cells towards the inside lighter, flattened, thick-walled. *Pseudoparaphyses* 2–3 µm wide, filamentous, branched septate. *Asci* 113–128 × 15–19 µm ($\bar{x}$ = 120 × 17 µm, *n* = 30), 8-spored, bitunicate, fissitunicate, clavate to cylindric-clavate, pedicellate, apex rounded with a minute ocular chamber. *Ascospores* 32–40 × 6–8 µm ($\bar{x}$ = 36 × 7 µm, *n* = 20), overlapping uni- to biseriate, narrowly fusiform, with a nearly median primary septum, deeply constricted at the septum, hyaline, guttulate, asymmetrical, conical, and narrowly rounded at the ends.

Culture characteristics: Ascospore germinating on PDA within 12 h and germ tubes produced from one end of the spore. Colonies on PDA, circular, reaching 6 cm in 45 days at room temperature, smooth surface, papillae, pale brown in PDA medium. Mycelium superficial, grayish-brown to brown, hairy, smooth, circular; reverse pale brown at the edges, dark brown in the middle, flocculent at the edge.

Material examined: China, Yunnan Province, Yuxi City, Jiangchuan District, Xingyunhu Lake, 24°23′05″ N, 102°48′22″ E (1720 m), on unknown submerged decaying wood, 10 July 2021, H.W. Shen, L511 (KUN-HKAS 124592), living culture (KUNCC 22–12417); ibid., 24°23′05″ N, 102°48′22″ E (1720 m), on the submerged stem of *Phragmites* sp. (*Poaceae*), 10 July 2021, S. Luan, L579 (KUN-HKAS 124596), living culture (KUNCC 22–12418); ibid., on submerged stem of *Phragmites* sp. (*Poaceae*), 10 July 2021, Y.K. Jiang, L474 (KUN-HKAS 124591), living culture (KUNCC 22–12416).

Known host and distribution: China, Guizhou Province, Anshun City, Gaodang Village, 26.071° N, 105.698° E, Suoluo River, on decaying bamboo culms submerged in a freshwater stream, HKAS 112629 (holotype) [15].

Notes: *Setoseptoria bambusae* was introduced by Yang et al. [15] to accommodate two collections, GZCC 17–0044 (ex-type strain) and IFRD500-013 (previously identified as *S. arundinaceae*, without description). In this study, our four new collections clustered with the ex-type strain of *S. bambusae* with 100% ML/1.00 PP statistical support (Figure 2). Furthermore, our collections fit the morphological characteristics of *S. bambusae* except for the size of asci and ascospores, our isolate has shorter asci (113–128 vs. 130–180 µm) and longer ascospores (32–40 vs. 28–37 µm). Therefore, we identified them as *S. bambusae*. Our four new collections were collected from lentic freshwater habitats. The holotype was collected from lotic habitats.

*Setoseptoria suae* H.W. Shen, K.D. Hyde & Z.L. Luo sp. nov. Figure 10.

MycoBank number: 849740; Facesoffungi number: FoF 14877.

Etymology: “suae” (Lat.) in memory of the Chinese mycologist Prof. Hong-Yan Su (4 April 1967–3 May 2022).

Holotype: KUN-HKAS 124595.

*Saprobic* on submerged decaying wood in a freshwater lake. Sexual morph: Undetermined. Asexual morph: Conidiomata 383–512 µm high, 173–196 µm wide, solitary, scattered, semi-immersed to immersed in the host, pycnidial, subglobose to ellipsoidal, unilocular, black, ostiolate, apapillate. *Ostiole* short, centrally located. *Conidiomatal wall* 33–55 µm wide, thickening at the upper zone, thick-walled, composed of several layers of *textura angularis,* an outer layer comprising brown to dark brown cells, pigmented; inner layer comprising hyaline cells. *Conidiophores* reduced to conidiogenous cells. *Conidiogenous cells* (4–)7–15(–26) × 4–6 µm ($\bar{x}$ = 11 × 5 µm, *n* = 25), arising from the inner layers of conidiomata, hyaline, enteroblastic, phialidic, determinate, ampuliform, subcylindrical to lageniform. Conidia 33–43 × 4–6 µm ($\bar{x}$ = 38 × 5 µm, *n* = 50), subcylindrical, with obtuse to subobtuse ends, straight or slightly curved, hyaline, (1–)3-septate, euseptate, mostly with one large central guttule per cell when young, with many small guttules in each cell at maturity, slightly constricted at the septum, smooth-walled.

Culture characteristics: Conidia germinated on PDA within 12 h and germ tubes produced from the ends of the spore. Colonies on PDA, circular, reaching 6 cm in one month at room temperature, brown to dark brown. Mycelium superficial, brown to dark brown, hairy, smooth, circular; dark brown from below.

Material examined: China, Yunnan Province, Yuxi City, Tonghai County, Qiluhuhu Lake, 24°08′37″ N, 102°46′24″ E (1800 m), on submerged stem of *Phragmites* sp. (Poaceae), 11 July 2021, H.W. Shen, L570 (KUN-HKAS 124595, holotype), ex-type living cultures (CGMCC 3.24266 = KUNCC 22–12419).

Notes: Phylogenetic analysis showed that *Setoseptoria suae* clustered with *S. phragmitis* with 100% ML/0.99 PP statistical support (Figure 2). The comparison of ITS and LSU sequences between *S. suae* and *S. phragmitis* shows that the similarities are 96.9% (538/555 bp) and 99.9% (826/827 bp), respectively. *Setoseptoria suae* resembles *S. phragmitis* in having immersed, globose conidiomata, hyaline, subcylindrical, smooth, guttulate, (1–)3-septate conidia [37]. However, *Setoseptoria suae* can be distinguished from *S. phragmitis* by its larger conidia (33–43 × 4–6 µm vs. (19–)25–35(–38) × (3.5–)4 µm). In addition, the conidia of *S. phragmitis* mostly have one large central guttule per cell, while *Setoseptoria suae* has conidia with many small guttules in each cell. We, therefore, introduce *S. suae* as a new species.

## 4. Discussion

Yunnan, located on the Yunnan–Guizhou Plateau, is one of the global biodiversity hotspots with rich biological resources [18,19,75]. In recent years, research on lignicolous freshwater fungi in Yunnan has developed rapidly, and a large number of new species have been reported from lotic freshwater habitats such as streams and rivers [10,13,76,77,78,79,80,81]. A few studies have reported lignicolous freshwater fungi from lentic habitats in Yunnan Province. For example, Cai et al. [17] and Luo et al. [2] investigated lignicolous freshwater fungi in Fuxianhu and Dianchi Lakes, respectively. However, freshwater fungi in lentic habitats have not been updated recently. In this study, we investigate the freshwater fungi in Cibihu, Luguhu, Qiluhu, Xihu, and Xingyunhu lakes in Yunnan Province, one new genus, two new species, and three new records are reported, the results indicate that high undiscovered diversity of lignicolous freshwater fungi in lentic habitats.

Zhang et al. [36] provided the first multigene phylogenetic analysis of Pleosporales and introduced the family Lentitheciaceae which accepted the genera *Lentithecium*, *Katumotoa,* and *Keissleriella*. Dong et al. [10] treated the family with ten genera and this was followed by Wijayawardene et al. [41]. Previous studies based on morphology and phylogenetic analyses showed that the classification of *Lentithecium*, *Keissleriella*, and *Setoseptoria* is confusing as the placement of several taxa was problematic and has been transferred to different genera. For example, Suetrong et al. [82] transferred *Keissleriella rara* to *Lentithecium* as *L. rarum*; however, later studies showed that *L. rarum* clustered with *K. trichophoricola* in *Keissleriella* [14]. Similarly, Zhang et al. [35] transferred *Keissleriella linearis* to *Lentithecium* as *L. lineare*, Singtripop et al. [83] re-examined the type specimen of *K. linearis* (*L. lineare*) and transferred it to *Keissleriella* based on LSU phylogenetic analysis, and this was confirmed by subsequent phylogenetic studies [14,34,72]. The placements of *Lentithecium* species have been revised in recent years based on multigene phylogenetic studies [14,34,84]. Calabon et al. [14] transferred several *Lentithecium* species with brown and versicolored ascospores without sheaths and hyaline conidia to *Halobyssothecium*, including *L. cangshanense*, *L. carbonneanum*, *L. kunmingense*, *L. unicellulare,* and *L. voraginesporum*. Currently, 13 species are accepted in *Halobyssothecium*. In the present study, we report the sexual morph of *H*. *phragmitis* and provide detailed morphological descriptions for its sexual morph.

## Figures and Tables

**Figure 1 jof-09-00962-f001:**
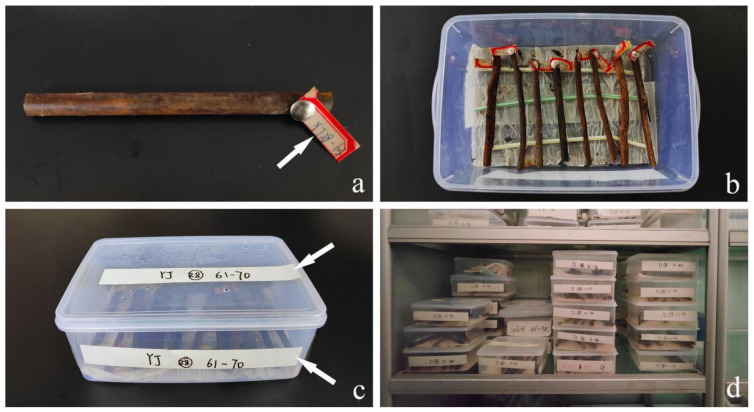
(**a**) Sample with a label (arrow indicates sample number); (**b**) Samples in the plastic box; (**c**) Plastic box with labels (arrows indicate labels documenting detailed sampling sites and sample order); (**d**) The samples were incubated on the culture rack.

**Figure 2 jof-09-00962-f002:**
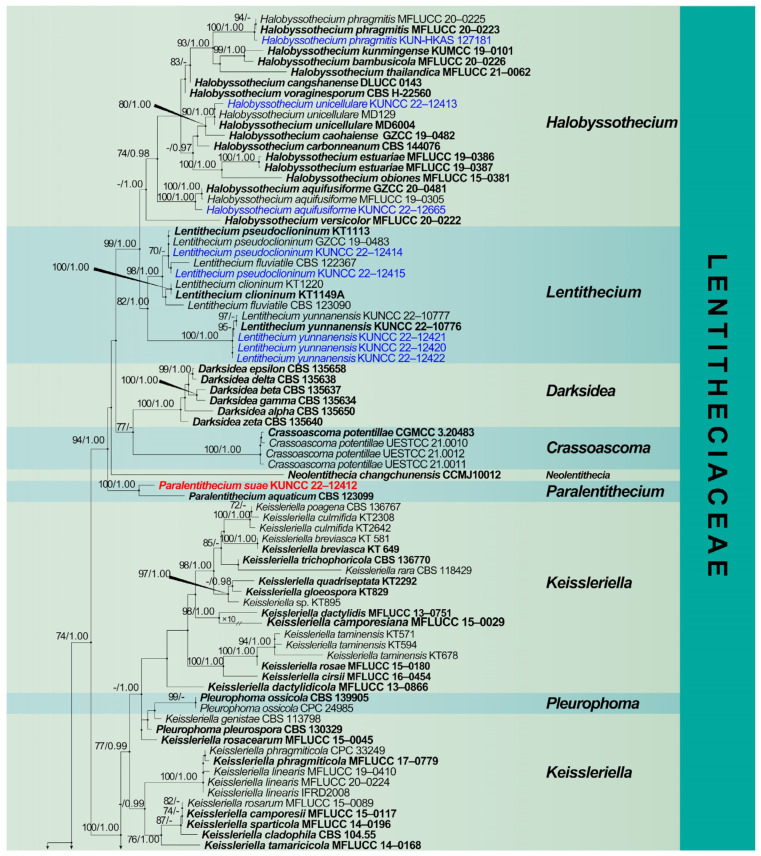
Maximum likelihood (ML) tree is based on combined LSU, SSU, ITS, and *tef 1-α* sequence data. Bootstrap support values with an ML greater than 70% and Bayesian posterior probabilities (PP) greater than 0.97 are given above the nodes, shown as “ML/PP”. The tree is rooted to *Pleomonodictys capensis* (CBS 968.97) and *P. descalsii* (CBS 142298). New species are indicated in red bold, new strains are indicated in blue, and type strains are in black bold.

**Figure 3 jof-09-00962-f003:**
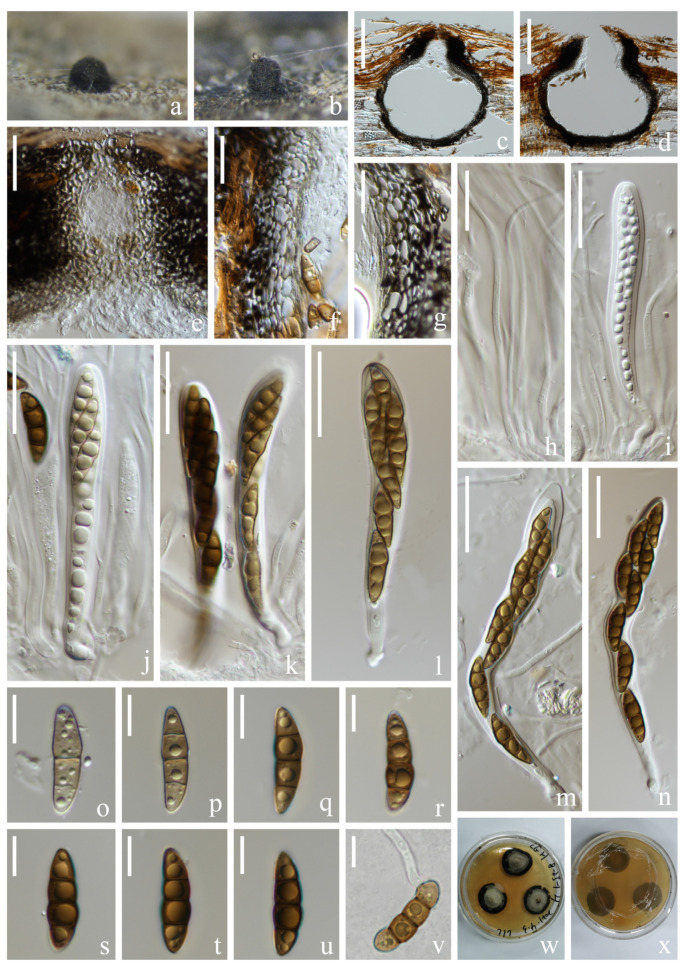
*Halobyssothecium aquifusiforme* (KUN-HKAS 1124599). (**a**,**b**) Appearance of ascomata on the host; (**c**,**d**) Sections of ascomata; (**e**) Ostiole; (**f**,**g**) Section of peridium; (**h**) Pseudoparaphyses; (**i**–**n**) Asci; (**o**–**u**) Ascospores; (**v**) Germinated conidium; (**w**,**x**) Colony on MEA, obverse (**w**) and reverse (**x**); Scale bar: (**c**,**d**) = 150 µm; (**e**,**i**–**n**) = 30 µm; (**f**–**h**) = 20 µm; (**o**–**v**) = 10 µm.

**Figure 4 jof-09-00962-f004:**
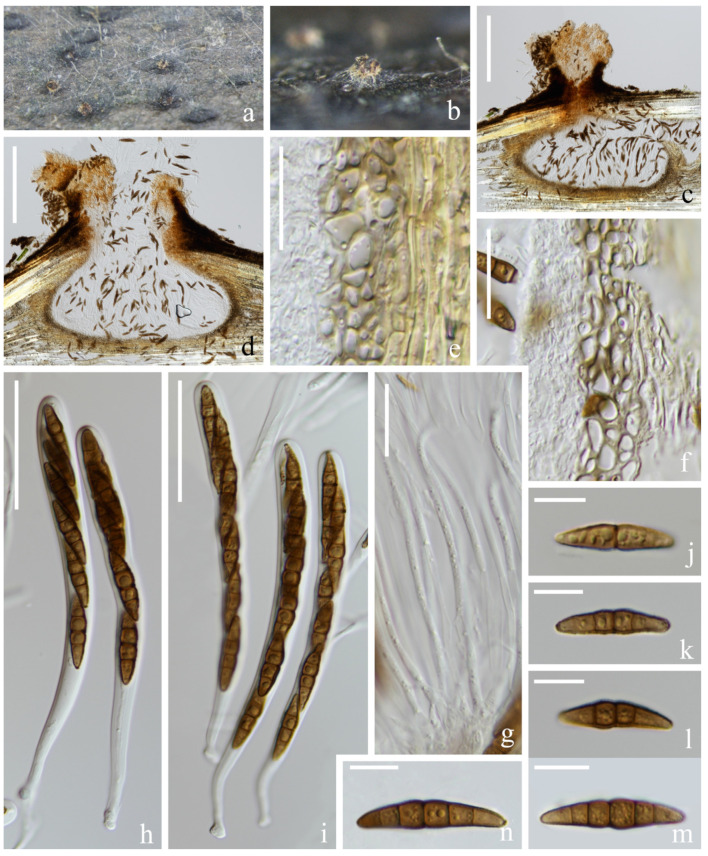
*Halobyssothecium phragmitis* (KUN-HKAS 124600, new geographic record). (**a**,**b**) Appearance of ascomata on the host; (**c**,**d**) Sections of ascomata; (**e**,**f**) Section of peridium; (**g**) Pseudoparaphyses; (**h**,**i**) Asci; (**j**–**n**) Ascospores. Scale bar: (**c**,**d**) = 200 µm; (**e**–**g**) = 20 µm; (**h**,**i**) = 40 µm; (**j**–**n**) = 10 µm.

**Figure 5 jof-09-00962-f005:**
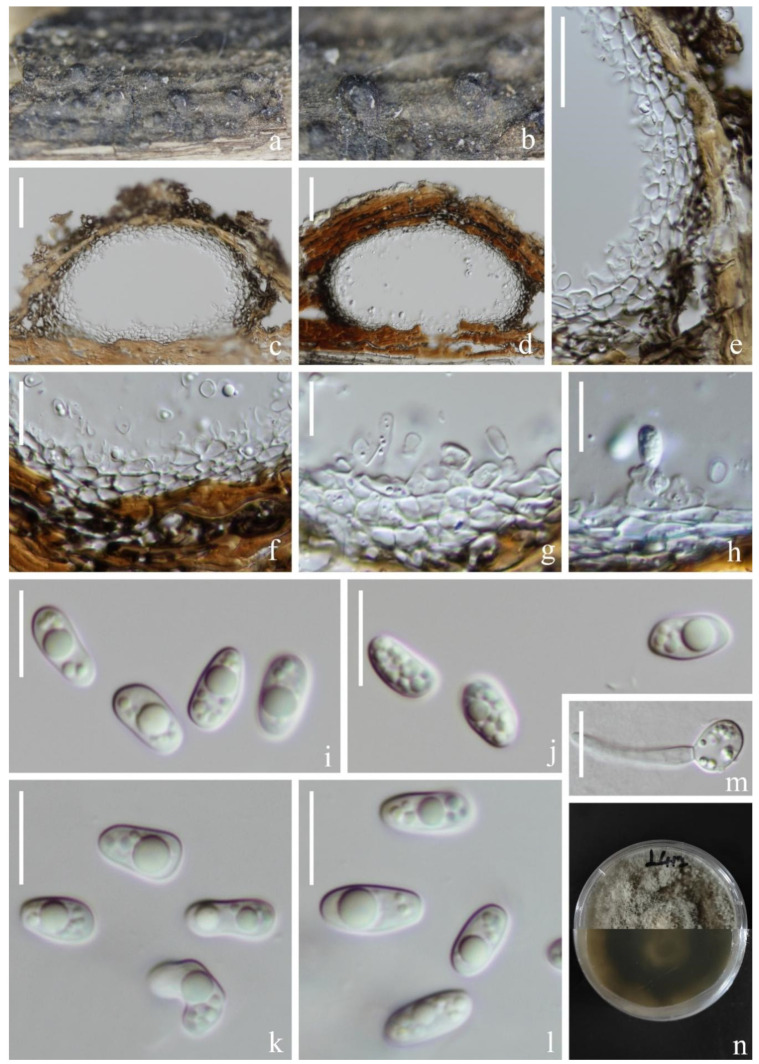
*Halobyssothecium unicellulare* (KUN-HKAS 124589, new geographic record). (**a**,**b**) Appearance of conidiomata on the host. (**c**,**d**) Sections of conidiomata. (**e**,**f**) Conidiomatal wall. (**g**,**h**) Developing conidia attach to conidiogenous cells. (**i**–**l**) Conidia. (**m**) Germinated conidium. (**n**) Colony on PDA, obverse (upper) and reverse (lower). Scale bar: (**c**,**d**) = 40 µm, (**e**,**f**) = 20 µm, (**g**–**m**) = 10 µm.

**Figure 6 jof-09-00962-f006:**
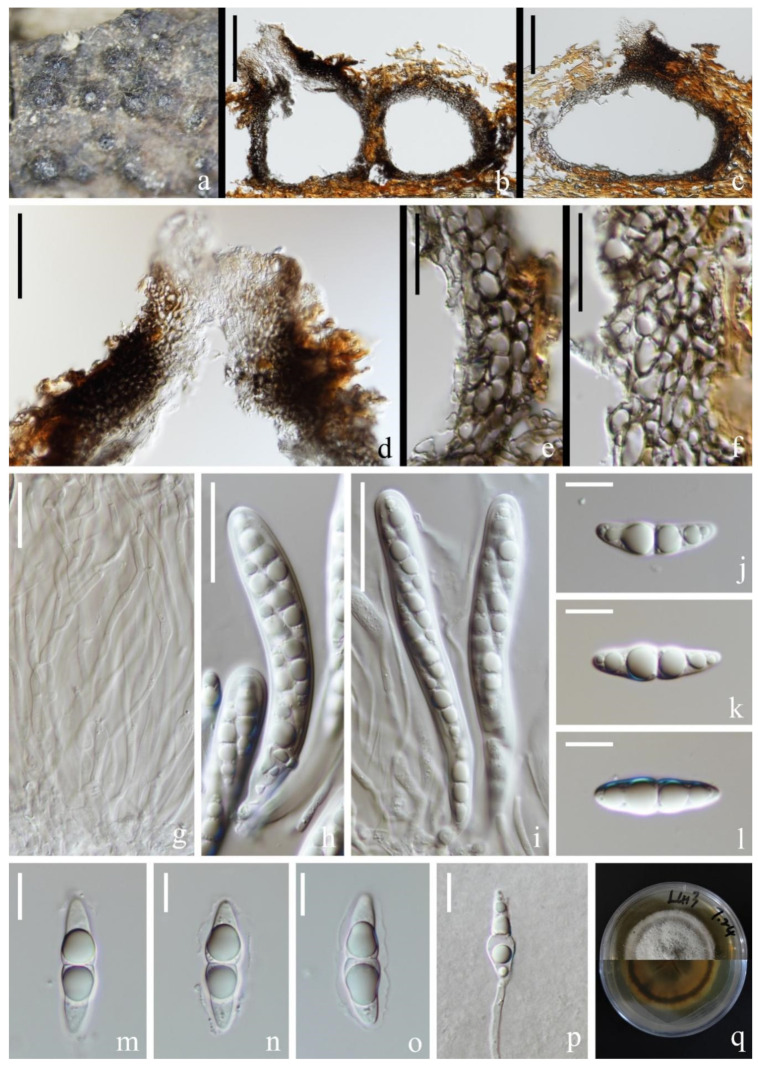
*Lentithecium pseudoclioninum* (KUN-HKAS 124590). (**a**) Appearance of ascomata on the host; (**b**,**c**) Sections of ascomata; (**d**) Ostiole; (**e**,**f**) Section of peridium; (**g**) Pseudoparaphyses; (**h**,**i**) Asci; (**j**–**o**) Ascospores; (**p**) Germinated conidium; (**q**) Colony on PDA, obverse (upper) and reverse (lower). Scale bar: (**b**,**c**) = 100 µm; (**d**) = 50 µm; (**e**–**g**) = 20 µm; (**h**,**i**) = 30 µm; (**j**–**p**) = 10 µm.

**Figure 7 jof-09-00962-f007:**
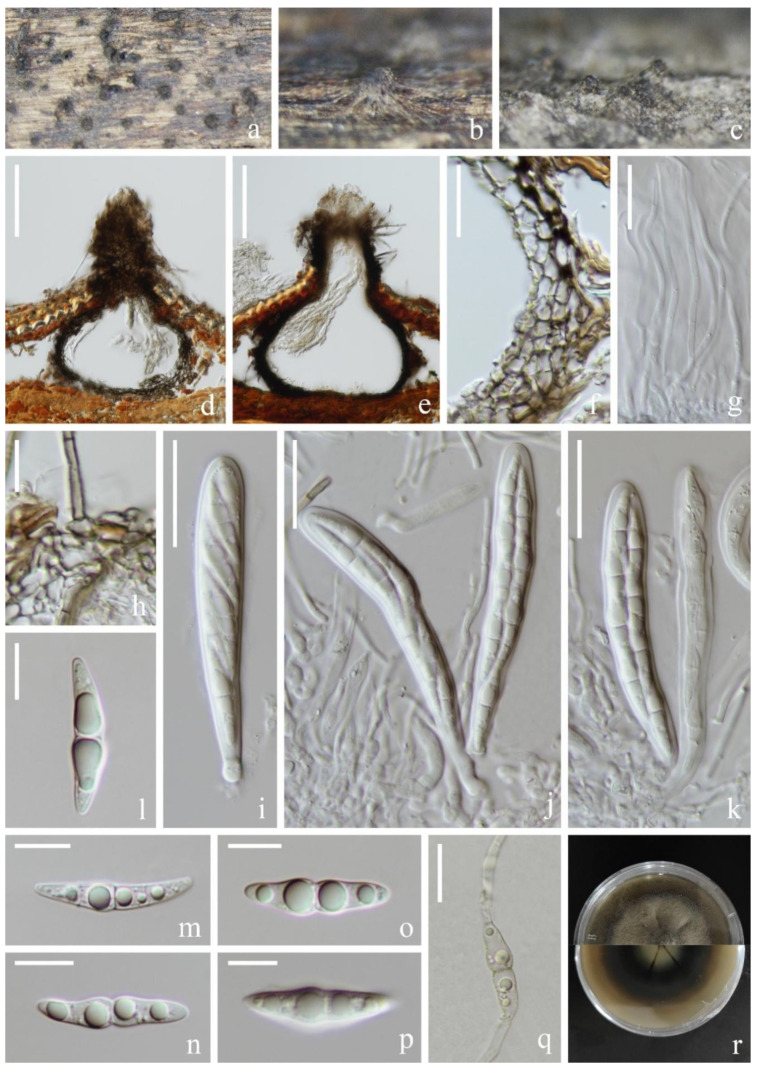
*Lentithecium yunnanensis* (KUN-HKAS 124597, new habitat records). (**a**–**c**) Appearance of ascomata on the host; (**d**,**e**) Sections of ascomata; (**f**) Section of peridium; (**g**) Pseudoparaphyses; (**h**) Ascomata wall with hypha; (**i**–**k**) Asci; (**l**–**p**) Ascospores; (**q**) Germinated conidium; (**r**) Colony on PDA, obverse (upper) and reverse (lower). Scale bar: (**d**,**e**) = 100 µm; (**f**,**g**) = 20 µm; (**i**–**k**) = 30 µm; (**h**,**l**–**q**) = 10 µm.

**Figure 8 jof-09-00962-f008:**
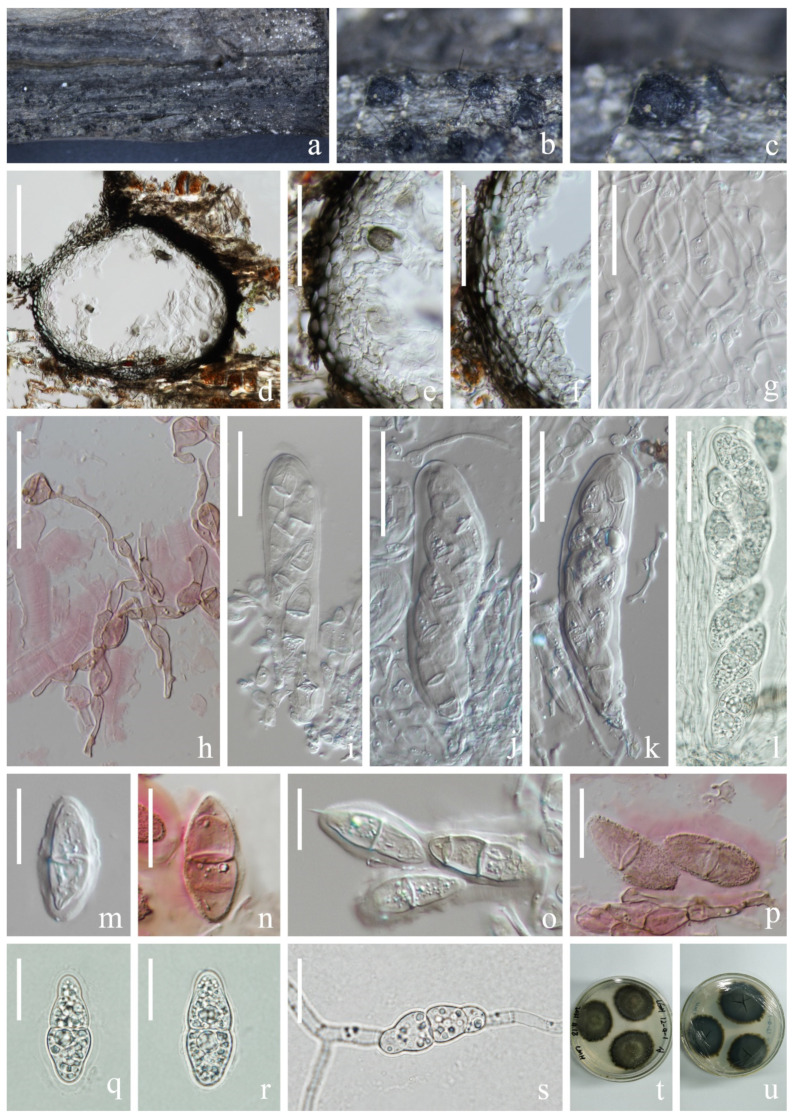
*Paralentithecium* suae (KUN-HKAS 124587, holotype). (**a**–**c**) Appearance of ascomata on the host; (**d**) Sections of ascomata; (**e**,**f**) Section of peridium; (**g**,**h**) Pseudoparaphyses; (**i**–**l**) Asci; (**m**–**r**) Ascospores; (**s**) Germinated conidium; (**t**,**u**) Colony on PDA, obverse (**t**) and reverse (**u**). Scale bar: (**d**) = 100 µm; (**e**–**l**) = 40 µm; (**m**–**s**) = 20 µm.

**Figure 9 jof-09-00962-f009:**
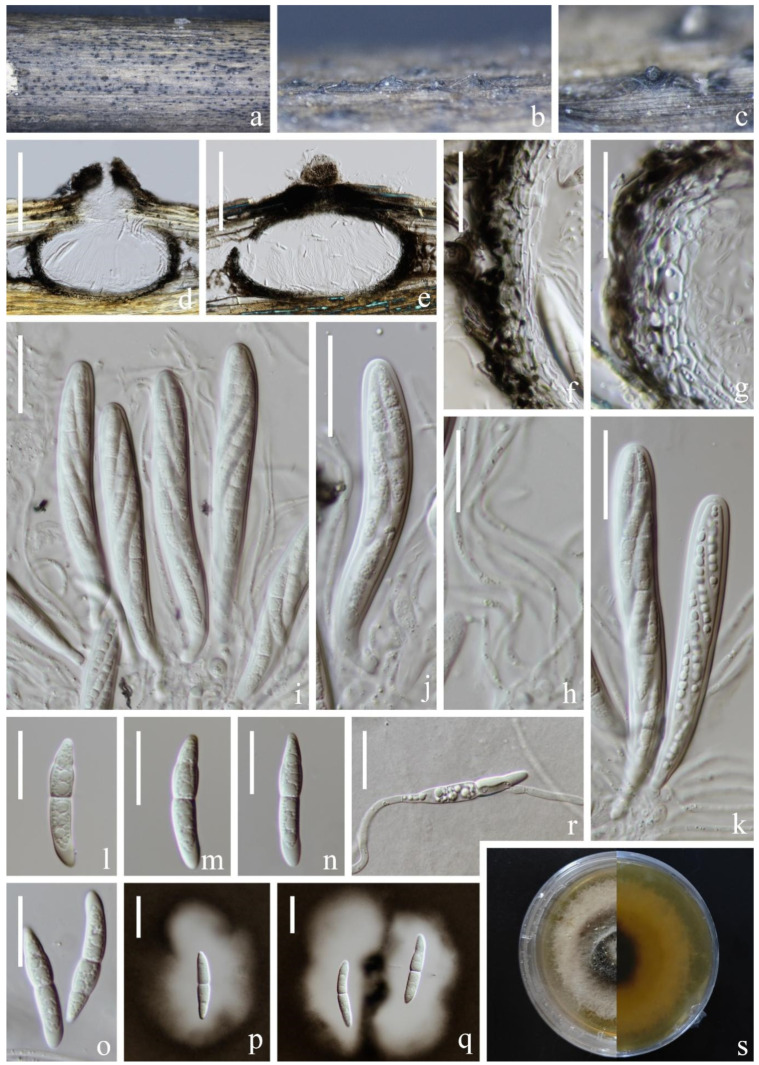
*Setoseptoria bambusae* (KUN-HKAS 124592). (**a**–**c**) Appearance of ascomata on the host; (**d**,**e**) Sections of ascomata; (**f**,**g**) Section of peridium; (**h**) Pseudoparaphyses; (**i**–**k**) Asci; (**l**–**o**) Ascospores; (**p**,**q**) Ascospore stained in Indian ink; (**r**) Germinated conidium; (**s**) Colony on PDA, obverse (**left**) and reverse (**right**). Scale bar: (**d**,**e**) = 150 µm; (**f**–**k**) = 30 µm; (**h**,**i**) = 30 µm; (**l**–**r**) = 20 µm.

**Figure 10 jof-09-00962-f010:**
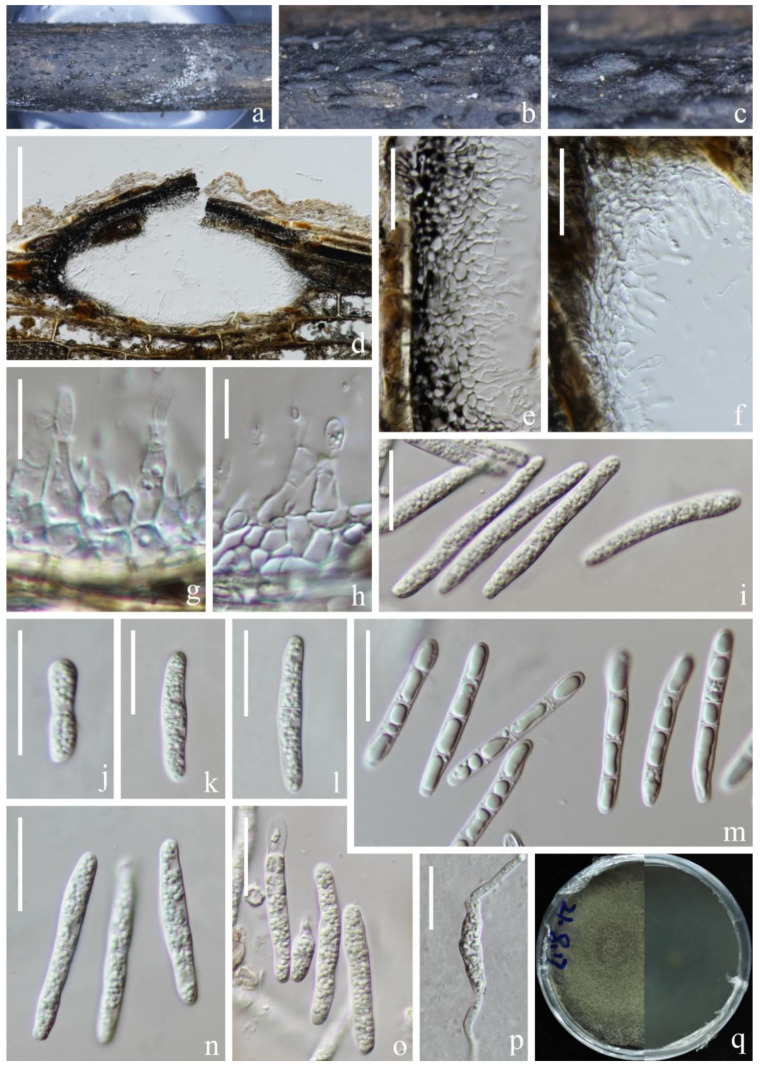
*Setoseptoria suae* (KUN-HKAS 124595, holotype). (**a**–**c**) Appearance of conidiomata on the host; (**d**) Sections of conidiomata; (**e**,**f**) Section of peridium; (**g**,**h**) Conidiomata and conidiogenous cells; (**i**–**o**) Conidia; (**p**) Germinated conidium; (**q**) Colony on PDA, obverse (**left**) and reverse (**right**). Scale bar: (**d**) = 100 µm; (**e**,**f**) = 30 µm; (**g**,**h**) = 10 µm; (**h**,**i**) = 30 µm; (**i**–**p**) = 10 µm.

**Table 1 jof-09-00962-t001:** Taxa used in the phylogenetic analyses and their corresponding GenBank accession numbers.

Species	Strain/Voucher Number	GenBank Accession Number			
		LSU	SSU	ITS	*tef 1-α*
*Bambusicola bambusae*	MFLUCC 11–0614 ^T^	JX442035	JX442039	NR_121546	KP761722
*Bambusicola irregulispora*	MFLUCC 11–0437 ^T^	JX442036	JX442040	NR_121547	KP761723
*Bambusicola massarinia*	MFLUCC 11–0389 ^T^	JX442037	JX442041	JX442033	KP761725
*Bambusicola splendida*	MFLUCC 11–0439 ^T^	JX442038	JX442042	NR121549	KP761726
*Crassoascoma potentillae*	UESTCC 21.0010	OK161254	OK161233	OK161237	OK181165
*Crassoascoma potentillae*	UESTCC 21.0011	OK161255	OK161234	OK161238	OK181166
*Crassoascoma potentillae*	UESTCC 21.0012	OK161256	OK161235	OK161239	OK181167
*Crassoascoma potentillae*	CGMCC 3.20483 ^T^	OK161257	OK161236	OK161240	OK181168
*Darksidea alpha*	CBS 135650 ^T^	KP184019	KP184049	NR_137619	KP184166
*Darksidea beta*	CBS 135637 ^T^	KP184023	KP184074	NR_137957	KP184189
*Darksidea delta*	CBS 135638 ^T^	KP184024	KP184069	NR_137075	KP184184
*Darksidea epsilon*	CBS 135658 ^T^	KP184029	KP184070	NR_137959	KP184186
*Darksidea gamma*	CBS 135634 ^T^	KP184031	KP184073	NR_137587	KP184188
*Darksidea zeta*	CBS 135640 ^T^	KP184013	KP184071	NR_137958	KP184191
*Halobyssothecium aquifusiforme*	GZCC 20–0481 ^T^	OP377925	OP378010	OP377825	OP473005
*Halobyssothecium aquifusiforme*	MFLUCC 19–0305	OP377929	OP378014	OP377829	OP473008
** *Halobyssothecium aquifusiforme* **	**KUNCC 22–12665**	**OR335346**	**OR335329**	**OR335289**	**OR367662**
*Halobyssothecium bambusicola*	MFLUCC 20–0226 ^T^	MT068489	MT068494	MN833419	MT477868
*Halobyssothecium cangshanense*	DLUCC 0143 ^T^	KU991149	KU991150	–	–
*Halobyssothecium caohaiense*	GZCC 19–0482 ^T^	MW133831	MW134611	OP377841	OP473019
*Halobyssothecium carbonneanum*	CBS 144076 ^T^	MH069699	–	MH062991	–
*Halobyssothecium estuariae*	MFLUCC 19–0386 ^T^	MN598871	MN598868	MN598890	MN597050
*Halobyssothecium estuariae*	MFLUCC 19–0387 ^T^	MN598872	MN598869	MN598891	MN597051
*Halobyssothecium kunmingense*	KUMCC 19–0101 ^T^	MN913732	MT864313	MT627715	MT954408
*Halobyssothecium obiones*	20AV2566	–	–	KX263862	–
*Halobyssothecium obiones*	27AV2385	–	–	KX263864	–
*Halobyssothecium obiones*	MFLUCC 15–0381 ^T^	MH376744	MH376745	MH377060	MH376746
*Halobyssothecium phragmitis*	MFLUCC 20–0223 ^T^	MT068486	MT068491	MT232435	MT477865
*Halobyssothecium phragmitis*	MFLUCC 20–0225	MT068488	MT068493	MT232437	MT477867
** *Halobyssothecium phragmitis* **	**HKAS 127181**	**OR506189**	**OR506192**	**OR506177**	**OR513794**
*Halobyssothecium thailandica*	MFLUCC 21–0062 ^T^	MZ433248	MZ429435	MZ429434	–
*Halobyssothecium unicellulare*	MD129	KX505375	KX505373	–	–
** *Halobyssothecium unicellulare* **	**KUNCC 22–12413**	**OR335347**	**OR335330**	**OR335290**	
*Halobyssothecium unicellulare*	MD6004 ^T^	KX505376	KX505374	–	–
*Halobyssothecium versicolor*	MFLUCC 20–0222 ^T^	MT068485	MW346047	MT232434	MT477864
*Halobyssothecium voraginesporum*	CBS H-22560 ^T^	KX499520	KX499519	–	–
*Kalmusia scabrispora*	KT2202	AB524594	AB524453	LC014576	AB539107
*Karstenula rhodostoma*	CBS 690.94	GU301821	GU296154	–	GU349067
*Katumotoa bambusicola*	KT 1517a ^T^	AB524595	AB524454	LC014560	AB539108
*Keissleriella bambusicola*	KUMCC 18–0122 ^T^	MK995880	MK995878	MK995881	MN213156
*Keissleriella breviasca*	KT 581	AB807587	AB797297	AB811454	AB808566
*Keissleriella breviasca*	KT 649 ^T^	AB807588	AB797298	AB811455	AB808567
*Keissleriella camporesiana*	MFLUCC 15–0029 ^T^	MN401741	MN401743	MN401745	MN397907
*Keissleriella camporesii*	MFLUCC 15–0117 ^T^	MN252886	MN252907	MN252879	–
*Keissleriella caraganae*	KUMCC 18–0164 ^T^	MK359439	MK359444	MK359434	MK359073
*Keissleriella cirsii*	MFLUCC 16–0454 ^T^	KY497780	KY497782	KY497783	KY497786
*Keissleriella cladophila*	CBS 104.55 ^T^	GU301822	GU296155	MH857391	GU349043
*Keissleriella culmifida*	KT2308	AB807591	AB797301	LC014561	AB808570
*Keissleriella culmifida*	KT2642	AB807592	AB797302	LC014562	AB808571
*Keissleriella dactylidicola*	MFLUCC 13–0866 ^T^	KT315506	KT315505	–	KT315507
*Keissleriella dactylidis*	MFLUCC 13–0751 ^T^	KP197668	KP197666	KP197667	KP197669
*Keissleriella genistae*	CBS 113798	GU205222	GU205242	–	–
*Keissleriella gloeospora*	KT829	AB807589	AB797299	LC014563	AB808568
*Keissleriella linearis*	IFRD2008	FJ795435	FJ795478	–	–
*Keissleriella linearis*	MFLUCC 19–0410	MN598873	MN598870	MN598892	MN607978
*Keissleriella linearis*	MFLUCC 20–0224	MT068487	MT068492	MT232436	MT477866
*Keissleriella phragmiticola*	CPC 33249	MT223903	–	MT223808	MT223715
*Keissleriella phragmiticola*	MFLUCC 17–0779 ^T^	MG829014	–	MG828904	–
*Keissleriella poagena*	CBS 136767	KJ869170	–	KJ869112	–
*Keissleriella quadriseptata*	KT2292 ^T^	AB807593	AB797303	AB811456	AB808572
*Keissleriella rara*	CBS 118429	GU479791	GU479757	–	–
*Keissleriella rosacearum*	MFLUCC 15–0045 ^T^	MG829015	MG829123	–	–
*Keissleriella rosae*	MFLUCC 15–0180 ^T^	MG829016	MG922549	–	–
*Keissleriella rosarum*	MFLUCC 15–0089 ^T^	MG829017	MG829124	MG828905	–
*Keissleriella sparticola*	MFLUCC 14–0196 ^T^	KP639571	–	–	–
*Keissleriella tamaricicola*	MFLUCC 14–0168 ^T^	KU900300	–	KU900328	–
*Keissleriella taminensis*	KT571	AB807595	AB797305	LC014564	AB808574
*Keissleriella taminensis*	KT594	AB807596	AB797306	–	–
*Keissleriella taminensis*	KT678	AB807597	AB797307	LC014565	AB808575
*Keissleriella trichophoricola*	CBS 136770 ^T^	KJ869171	–	KJ869113	–
*Keissleriella yonaguniensis*	HHUF 30138 ^T^	AB807594	AB797304	AB811457	AB808573
*Keissleriella* sp.	KT895	AB807590	AB797300	–	AB808569
*Latorua caligans*	CBS 576.65 ^T^	MH870362	–	MH858723	–
*Latorua grootfonteinensis*	CBS 369.72 ^T^	MH877741	–	–	–
*Lentithecium clioninum*	KT1149A ^T^	AB807540	AB797250	LC014566	AB808515
*Lentithecium clioninum*	KT1220	AB807541	AB797251	LC014567	AB808516
*Lentithecium fluviatile*	CBS 122367	FJ795451	FJ795493	–	GU349074
*Lentithecium fluviatile*	CBS 123090	FJ795450	FJ795492	–	–
*Lentithecium pseudoclioninum*	KT1113 T	AB807544	AB797254	AB809632	AB808520
*Lentithecium pseudoclioninum*	GZCC 19–0483	MW133832	MW134612	OM692194	–
** *Lentithecium pseudoclioninum* **	**KUNCC 22–12414**	**OR335348**	**OR335331**	**OR335291**	–
** *Lentithecium pseudoclioninum* **	**KUNCC 22–12415**	**OR335349**	**OR335331**	**OR335291**	–
*Lentithecium yunnanensis*	KUNCC 22–10776 ^T^	ON227127	ON227123	ON227126	ON228074
*Lentithecium yunnanensis*	KUNCC 22–10777	ON227124	ON227122	ON227125	ON228075
** *Lentithecium yunnanensis* **	**KUNCC 22–12420**	**OR335350**	**OR335333**	**OR335293**	OR367664
** *Lentithecium yunnanensis* **	**KUNCC 22–12421**	**OR335351**	**OR335334**	**OR335294**	OR367665
** *Lentithecium yunnanensis* **	**KUNCC 22–12422**	**OR335352**	**OR335335**	**OR335295**	OR367666
*Longipedicellata aptrootii*	MFLUCC 10–0297 ^T^	KU238894	KU238895	KU238893	KU238892
*Longipedicellata aptrootii*	MFLUCC 18–0988	MN913744	–	MT627733	–
*Macrodiplodiopsis desmazieri*	CBS 140062 ^T^	KR873272	–	KR873240	–
*Massarina cisti*	CBS 266.62	FJ795447	FJ795490	LC014568	AB808514
*Massarina eburnea*	CBS 139697	AB521735	AB521718	LC014569	AB808517
*Massarina eburnea*	CBS 473.64	GU301840	GU296170	AF383959	GU349040
*Multiseptospora thailandica*	MFLUCC 11–0183 ^T^	KP744490	KP753955	KP744447	KU705657
*Murilentithecium clematidis*	MFLUCC 14–0561	KM408758	KM408760	KM408756	KM454444
*Murilentithecium clematidis*	MFLUCC 14–0562 ^T^	KM408759	KM408761	KM408757	KM454445
*Murilentithecium lonicerae*	MFLUCC 18–0675 ^T^	MK214373	MK214376	MK214370	MK214379
*Murilentithecium rosae*	MFLUCC 15–0044 ^T^	MG829030	MG829137	MG828920	–
*Neolentithecia changchunensis*	CCMJ10012 ^T^	MZ518790	MZ518820	MZ519071	–
*Neoophiosphaerella sasicola*	KT1706 ^T^	AB524599	AB524458	LC014577	AB539111
*Parabambusicola thysanolaenae*	KUMCC 18–0147 ^T^	MK098199	MK098205	MK098190	MK098209
*Parabambusicola thysanolaenae*	KUMCC 18–0148	MK098198	MK098202	MK098193	MK098211
*Paraconiothyrium brasiliense*	CBS 100299 ^T^	JX496124	AY642523	JX496011	–
*Paraphaeosphaeria michotii*	MFLUCC 13–0349 ^T^	KJ939282	KJ939285	KJ939279	–
*Paraphaeosphaeria minitans*	CBS 122788	EU754173	EU754074	–	GU349083
*Phragmocamarosporium hederae*	MFLUCC 13–0552 ^T^	KP842915	KP842918	–	–
*Phragmocamarosporium platani*	MFLUCC 14–1191 ^T^	KP842916	KP842919	–	–
*Phragmocamarosporium rosae*	MFLUCC 17–0797 ^T^	MG829051	MG829156	–	MG829225
*Pleomonodictys descalsii*	CBS 142298 ^T^	KY853522	–	KY853461	–
*Pleomonodictys capensis*	CBS 968.97 ^T^	KY853521	–	KY853460	–
*Pleurophoma ossicola*	CBS139905 ^T^	KR476769	–	KR476736	–
*Pleurophoma ossicola*	CPC24985	KR476770	–	KR476737	–
*Pleurophoma pleurospora*	CBS130329 ^T^	JF740327	–	–	–
*Poaceascoma aquaticum*	MFLUCC 14–0048 ^T^	KT324690	KT324691	–	–
*Poaceascoma halophila*	MFLUCC 15–0949 ^T^	MF615399	MF615400	–	–
*Poaceascoma helicoides*	MFLUCC 11–0136 ^T^	KP998462	KP998463	KP998459	KP998461
*Poaceascoma taiwanense*	MFLUCC 18–0083 ^T^	MG831567	MG831568	MG831569	–
*Paralentithecium aquaticum*	CBS 123099 ^T^	GU301823	GU296156	NR_160229	GU349068
** *Paralentithecium suae* **	**CGMCC 3.24265 ^T^**	**OQ732683**	**OQ875040**	**OQ874972**	**OR367672**
*Pseudokeissleriella bambusicola*	CGMCC 3.20950 ^T^	ON614138	ON614096	ON614135	ON639623
*Pseudokeissleriella bambusicola*	UESTCC 22.0028	ON614137	ON614095	ON614134	ON639622
*Setoseptoria arundelensis*	MFLUCC 17–0759 ^T^	MG829073	MG829173	MG828962	–
*Setoseptoria arundinacea*	CBS 123131	GU456320	GU456298	–	GU456281
*Setoseptoria arundinacea*	CBS 619.86	GU301824	GU296157	–	–
*Setoseptoria arundinacea*	MAFF 239460	AB807574	AB797284	LC014594	AB808550
*Setoseptoria arundinacea*	MAFF 243842 ^T^	AB807575	AB797285	LC014595	AB808551
*Setoseptoria bambusae*	GZCC 17–0044	OP377919	OP378004	OP377820	OP472999
** *Setoseptoria bambusae* **	**KUNCC 22–12416**	**OR335353**	**OR335336**	**OR335296**	**OR367667**
** *Setoseptoria bambusae* **	**KUNCC 22–12417**	**OR335354**	**OR335337**	**OR335297**	**OR367668**
** *Setoseptoria bambusae* **	**KUNCC 22–12418**	**OR335355**	**OR335338**	**OR335298**	**OR367669**
*Setoseptoria englandensis*	MFLUCC 17–0778 ^T^	MG829074	MG829174	MG828963	–
*Setoseptoria lulworthcovensis*	MFLU 18–0110 ^T^	MG829075	MG829175	–	–
*Setoseptoria magniarundinacea*	KT1174	AB807576	AB797286	LC014596	AB808552
*Setoseptoria phragmitis*	CBS 114802 ^T^	KF251752	–	KF251249	KF253199
*Setoseptoria phragmitis*	CBS 114966	KF251753	–	KF251250	KF253200
*Setoseptoria scirpi*	MFLUCC 14–0811 ^T^	KY770982	KY770980	MF939637	KY770981
** *Setoseptoria suae* **	**CGMCC 3.24266 ^T^**	**OQ874972**	**OQ875041**	**OQ874972**	**OR367673**
*Splanchnonema platani*	CBS 221.37	MH867404	–	MH855894	DQ677908
*Splanchnonema platani*	CBS 222.37	KR909316	KR909318	KR909311	KR909319
*Tingoldiago clavata*	MFLUCC 19–0495	MN857180	MN857188	MN857184	–
*Tingoldiago clavata*	MFLUCC 19–0496 ^T^	MN857178	MN857186	MN857182	–
*Tingoldiago clavata*	MFLUCC 19–0498	MN857179	MN857187	MN857183	–
*Tingoldiago graminicola*	KH155	AB521745	AB521728	LC014599	AB808562
*Tingoldiago graminicola*	KH68 ^T^	AB521743	AB521726	LC014598	AB808561
*Tingoldiago graminicola*	KT891	AB521744	AB521727	LC014600	AB808563
*Tingoldiago hydei*	MFLUCC 19–0499 ^T^	MN857177	–	MN857181	–
*Towyspora aestuari*	MFLUCC 15–1274 ^T^	KU248852	KU248853	NR_148095	–

Notes: The ex-type cultures are indicated using “T” after strain numbers; newly generated sequences are indicated in bold. “–” stands for no sequence data in GenBank.

## Data Availability

All sequences generated in this study were submitted to GenBank database.

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
