# Peer review of "Lignicolous Freshwater Fungi from Plateau Lakes in China (I): Morphological and Phylogenetic Analyses Reveal Eight Species of Lentitheciaceae, Including New Genus, New Species and New Records"

_jof, 2023, doi:10.3390/jof9100962_

Round 1
Reviewer 1 Report
A review on ms entitled “Lignicolous freshwater fungi from plateau lakes in China (І): Morphological and phylogenetic analyses reveal eight species of Lentitheciaceae, including new genera, new species and new records.” Hong-Wei Shen et al.
In my opinion, the manuscript/study possess a well written Introduction, wonderful mycological description (excellent desriptive illustrations-photos work here), and well-done Discussion. It is a really good mycology here!
However, my main concern goes to phylogeny as follows:
Data for Phylogeny Table 1
1. Halobyssothecium obiones 20AV2566 and 27AV2385, Halobyssothecium phragmitis HKAS 127181, (skip them, as they do not have complete dataset, plus you have a type culture for each included anyway)
2. For a type culture of Paralentithecium suae CGMCC 3.24265TGenBank numbers for LSU OQ732683 as well as ITS OQ874971 give reference to another strain KUNCC:22-12671 (“Kirschsteiniothelia sp.”), and I do not why? Please correct or explain because your type strain should be (CGMCC 3.24265 = KUNCC 22–12412).
3. Typo-mistakes in typing Paralentithecium (it is “ Pralentithecium aquaticum and “Pralentithecium suae”
4. Main concern here: your dataset for tef is definitely incomplete, not only that you use gaps from previous studies (missing sequences) but you even produce next gaps by not including tef sequences for your own species! Who should do this if not the authors who include/introduce novel taxa? You can not simply claim it is multi-loci analysis including this protein coding gene, when it is simply not true here. Unfortunately, 42% of tef sequences are missing (including those for new species). Thus, I am afraid to say, you cannot construct a relevant phylogenetic tree with such an incomplete dataset.
There is no doubt, your new species are clearly distant in ribosomal DNA loci from the close relatives, solid distinction in ITS indicates novelty. They are new! Well, I would suggest (a) either to skip tef and construct ITS and LSU concatenated (perhaps with SSU too), or (b) re-do your study and include all missing tef sequences (that would mean to make new PCR from type material if available at least for most crucial strains/species).
5. Figure 2 Phylotree. Please, again, not concatenation with tef (incomplete) use only those you have available (some gaps might be acceptable in very low minority) here is missing lot...
6. Figure 2 Phylotree. If it is printed the letters are too small for the readers, so please make it easier to read (divide in two pages)
7. Figure 2 Phylotree. Your records are in blue (not only the new species), for the new species use another color!
In the title Line 4 it is “ new genera” , should be “ new genus” !
----end---
Reviewer 2 Report
Dear Authors
I have had the opportunity to review your manuscript titled "Lignicolous freshwater fungi from plateau lakes in China (І): Morphological and phylogenetic analyses reveal eight species of Lentitheciaceae, including new genera, new species, and new records." Overall, I appreciate the effort you've put into this research, and I believe your study holds significant potential. However, there are a few areas that require attention before the manuscript can be considered ready for publication. I have listed the key suggestions below; the minor ones are listed in the attached PDF.
1. The individual single gene datasets should be analysed using both phylogenetic methods, and the outcomes should be included in the result. All the single gene trees should be added to the manuscript as supplements.
2. Both single gene and concetenated alignments and the respective trees should be submitted to a repository, such as TreeBase, and the accession number should be added to the text.
3. Both the new fungal species described in this manuscript are based on single isolates. Describing fungal species based on a single isolate can have several merits within the context of mycology and scientific research. While it is a topic of ongoing debate and consideration, there are valid reasons supporting this approach; for example, many fungal species are not easily isolated. Hence, this should be carefully considered and guided by the specific circumstances and available resources. So, it would be important for me to check the single gene trees before making a call. For now, unfortunately, I don't agree with it.

Some minor edits are required nothing serious.
Author Response
Please see the attachment, thank you.

Round 2
Reviewer 1 Report
If the dataset of tef (GenBank numbers) for new species, as claimed by the authors are in the Table and used for phyl. construction, I have no doubt of a substantial improvement here, thus this manuscript is suitable for JOF publication.
thank you for following my comments/recommendation.
kind regards RL
Reviewer 2 Report
Dear Authors,
Thanks for addressing my comments on the previous version of the manuscript. I also thank the authors for submitting the single gene trees. I agree with taxonomy of the two new species, Setoseptoria suae and Paralentithecium suae based on single isolates. However, when checking the single gene trees, I noticed that Paralentithecium suae (KUNCC 22-12414) has been labelled as "Lentithecium pseudoclioninum KUNCC 22–12414" in the ITS tree – this is incorrect. Therefore, I am requesting the authors to please ensure the taxon names are consistent across all trees and also in the alignment and trees submitted to the TreeBase.
Best regards,